# Prior Infection by *Colletotrichum spinaciae* Lowers the Susceptibility to Infection by Powdery Mildew in Common Vetch

**DOI:** 10.3390/plants13010052

**Published:** 2023-12-22

**Authors:** Faxi Li, Rui Zhu, Feng Gao, Tingyu Duan

**Affiliations:** 1State Key Laboratory of Herbage Improvement and Grassland Agro-ecosystems, Key Laboratory of Grassland Livestock Industry Innovation, Ministry of Agriculture and Rural Affairs, College of Pastoral Agriculture Science and Technology, Lanzhou University, Lanzhou 730000, China; lifx19@lzu.edu.cn (F.L.); 220220902410@lzu.edu.cn (R.Z.); 2Gansu Vocational College of Agriculture, Lanzhou 730020, China

**Keywords:** common vetch, anthracnose, powdery mildew, reactive oxygen species, defense enzyme

## Abstract

Anthracnose (*Colletotrichum spinaciae*) and powdery mildew (*Erysiphe pisi*) are important diseases of common vetch (*Vicia sativa*) and often co-occur in the same plant. Here, we evaluate how *C. spinaciae* infection affects susceptibility to *E. pisi*, using sterilized and non-sterilized field soil to test the effect of resident soil microorganisms on the plant’s immune response. Plants infected with *C. spinaciae* (C^+^) exhibited a respective 41.77~44.16% and 72.37~75.27% lower incidence and severity of powdery mildew than uninfected (C^−^) plants. Moreover, the net photosynthetic rate, transpiration rate, and stomatal conductance were higher in the C^−^ plants than in the C^+^ plants prior to infection with powdery mildew. These differences were not recorded following powdery mildew infection. Additionally, the activities of superoxide dismutase, polyphenol oxidase, and catalase were higher in the C^+^ plants than in the C^−^ plants. The resident soil microbiota did not affect the plant responses to both pathogens. By uncovering the mechanistic basis of plant immune response, our study informs integrated disease management in a globally important forage crop.

## 1. Introduction

Common vetch (*Vicia sativa*) is a member of the Leguminosae family. It originated from southern Europe and western Asia and is widely cultivated throughout the world [1]. Common vetch is an excellent forage crop that is highly resistant to drought, tolerant to salt, and highly effective at fixing nitrogen. Moreover, it grows quickly and can improve the productivity of pastures when mixed with other forage grasses [2]. Common vetch is a green manure crop that is commonly used in rotation and intercropping production, such as intercropping with fruit trees [3]. Moreover, the plants have been planted to prevent soil erosion [4].

Plant diseases significantly affect agricultural production both directly and indirectly and lead to losses of approximately $40 billion per year worldwide [5]. Common vetch is susceptible to diseases and pathogens, including fungi, bacteria, viruses, and nematodes. Approximately 24 fungal diseases, four bacterial diseases, five viral diseases, and 24 nematodes, including southern root-knot nematodes, have been detected in common vetch [6,7]. However, anthracnose is one of the most common diseases of common vetch. The pathogens that cause anthracnose in this plant include *Colletotrichum vicia* [8], *C. villosum* [9], *C. sativum* [10], *C. viciae-sativae* [11], *C. acutatum* [12], *C. lentis* [13], and *C. spinaciae* [14]. *Colletotrichum* species are hemibiotrophic plant pathogenic fungi that feed on live plant tissues in the early stage of infection and dead cells in the plant residues after plant death [15,16]. Anthracnose can cause severe yield losses. For example, it can decrease the plant biomass by 26.90~40.00%, decrease the numbers of effective nodules by 42.25%, and increase the content of jasmonic acid (JA) by 70.34% under greenhouse conditions [17]. Moreover, anthracnose reduces the content of chlorophyll a and b and inhibits the PSII photochemical activity [18]. *Colletotrichum spinaciae* was detected in Xiahe, Gansu Province, China, in 2019, and it caused a significant reduction in yield from 2019 to 2021. Anthracnose can reduce the shoots and seed yield of common vetch by more than 25.8% and 32.5%, respectively, under field conditions [19].

Powdery mildew is a common fungal disease that affects many plant species, including the *Erysiphe pisi* infection of common vetch. This disease decreases the ability to conduct photosynthesis, which results in severe yield losses [20,21,22]. *Erysiphe pisi* is a type of biotrophic parasitic pathogenic fungus that coexists with the host and does not cause its death [16,20,23]. Powdery mildew usually affects the photosynthesis of plants, which leads to yield losses. For example, powdery mildew can cause approximately 20% yield losses in winter wheat (*Triticum aestivum*) [24]. Powdery mildew can significantly reduce the net photosynthesis and evapotranspiration compared to in leaves that are not infected. Furthermore, the net photosynthesis can be practically interrupted when the infected area is more than 75% of a young leaf [25]. The induction in activities of enzymes related to defense, such as phenylalanine ammonia lyase, polyphenol oxidase (PPO), and peroxidase (POD), can decrease the natural incidence of powdery mildew infection by 37.2% [26].

Multiple stresses, such as abiotic ones, diseases, and insect pests, simultaneously affect crops in the agricultural system. Moreover, many studies have assessed the interaction between abiotic stress and disease. Most cases have shown that abiotic stress aggravates diseases. For example, salt and water stresses aggravate the susceptibility of faba bean (*Vicia faba*) to phytopathogenic fungi [27]. Moreover, banana (*Musa nana*) and grapevine (*Vitis vinifera*) are more vulnerable to diseases and infections under water stress [28,29]. Furthermore, some studies have analyzed the interactions between pathogens and insects and found that insect pests may positively or negatively affect the prevalence of diseases [19]. In some cases, a pathogenic infection can induce a plant immune response, which renders the plant resistant to the invasion of other pathogens. For example, infection with Zucchini yellow mosaic virus (ZYMV) reduces the exposure of a second insect-vectored pathogen (*Erwinia tracheiphila*) and the establishment of powdery mildew on *Cucurbita pepo* [30]. Additionally, infection with *Blumeria graminis* can elicit immune responses in wheat (*Triticum aestivum*), such as the expression of defense genes (*PAL*, *PR1*, *PR2*, *PR3*, *PEROX*, and *NADPHOX*), the content of total phenolics, and the activity of related antioxidant enzymes, thus suppressing the development of the English grain aphid (*Sitobion avenae*) [31]. Scientists have defined these pre-infection interactions as “ecological interference,” which indicates that a primary infection can reduce the rate of contact and development of secondary infections [32]. The previous infection in plant cells can trigger signaling cascades whose functions are implemented through the signaling molecules JA and salicylic acid (SA). The elicited immune system can provide broad-spectrum immunity against subsequent pathogen attacks [33]. However, the interaction between phytopathogenic fungi on forage crops is unclear.

Microbes in the rhizosphere can protect plants against abiotic and biotic stresses. This microbial resource has increasing importance for crop production and a reduction in fertilizer and pesticide inputs [34]. The protection of plants against diseases is one of the most intensively studied microbiome functions and became well known from the discovery of disease-suppressive soils [35]. For example, infection by the foliar downy mildew pathogen *Hyaloperonospora arabidopsidis* promotes the growth of specific microbiota in the rhizosphere of *Arabidopsis thaliana*, which indirectly affects the pathogen. These rhizosphere microbial community changes include an enrichment of three beneficial bacteria (*Xanthomonas*, *Stenotrophomonas*, and *Microbacterium*) that help plants to resist subsequent invasion by pathogens [36].

Anthracnose and powdery mildew are common diseases of common vetch that significantly affect the growth and production of this species in the Qinghai–Tibet Plateau in China. Infection with anthracnose usually begins in July, while powdery mildew occurs in September in this plateau. The two diseases simultaneously occur in middle and late September in the fields of the Qinghai–Tibet Plateau. Therefore, it is necessary to determine the synergistic effect of the two diseases on common vetch and assess whether anthracnose affects subsequent infection by powdery mildew. This study was designed to simulate a field anthracnose infection followed by that of powdery mildew to test the effects of anthracnose caused by *C. spinaciae* on a subsequent powdery mildew infection. We established different soil microbial treatments through the sterilization and non-sterilization of field soil with common vetch growth and determined the effects of the resident soil microorganisms in the field on the response of common vetch to anthracnose and powdery mildew. We proposed the following hypotheses: (1) The synergistic occurrence of anthracnose and powdery mildew severely damages common vetch, and anthracnose renders common vetch non-susceptible to powdery mildew, and (2) the resident field soil microorganisms have a positive effect on the resistance of common vetch to anthracnose and powdery mildew.

## 2. Materials and Methods

### 2.1. Plants and Pathogens

*Vicia sativa* “Lanjian No. 2” was obtained from Lanzhou University (Lanzhou, China). This is the primary common vetch cultivar that is recommended by the Ministry of Agriculture and Rural Affairs of China and widely planted in the Qinghai–Tibet Plateau. The anthracnose pathogen was isolated from infected leaves of common vetch in Xiahe County, Gansu Province. The pathogen was isolated from the diseased common vetch in the field and identified as *C. spinaciae* using morphological, molecular biological, and pathogenicity determination [14]. Powdery mildew occurs naturally in the individual plant pathology greenhouse.

Experimental design. The study was divided into two stages to simulate the mixed infection of anthracnose (occurring annually from July to September) and powdery mildew (occurring annually from August to October) in the greenhouse.

Stage 1: Stage 1 was designed to establish the anthracnose pathogen *C. spinaciae*-infected (C^+^) and uninfected (C^−^) plants for stage 2. In this step, common vetch was either infected or uninfected with *C. spinaciae* (30 pots for each treatment). The incidence and disease index (DI) of anthracnose were assessed to determine the successful colonization of *C. spinaciae* on the leaves of common vetch plants.

Stage 2: A fully crossed two-factor experiment was designed, which included *C. spinaciae* (two levels) × soil (two levels) = four treatments, each with 10 replicate pots. The disease incidence and DI of powdery mildew and the physiological and biochemical indices of common vetch were determined after the spontaneous occurrence of powdery mildew.

### 2.2. Growth Medium

The common vetch was cultivated in two soils, including field soil (FS) with common vetch, taken from the trial field in Xiahe County, and sterilized field soil (SFS) to establish growth media with resident field soil microorganisms and exclude resident field soil microorganisms, respectively. The soil mixtures were prepared by mixing 25% soil with 75% sand, which was then passed through a 2 mm sieve. The SFS and sands were sterilized using autoclaving twice at 121 °C for 1 h for 3 d and then dried in an oven at 110 °C for 36 h. The pots measured 14 cm (bottom diameter) by 22 cm (height) by 17 cm (top diameter) and contained 2.5 kg of soil.

### 2.3. Experiment Establishment, Management, and Harvest

Stage 1. The seeds of “Lanjian No. 2” were surface-sterilized for 10 min using 10% sodium hypochlorite (NaOCl), and then washed three times with sterilized distilled water. The treated seeds were placed in sterile Petri dishes that contained sterile filter paper at the bottom and incubated at 25 °C for 48 h to germinate. A total of 12 germinated seeds were planted in each pot and thinned to eight seedlings of a similar size 1 week after emergence for further experiments.

The *Colletotrichum spinaciae* was grown on potato dextrose agar (PDA). The conidia of the pathogen were harvested in sterilized water from 30-day-old cultures that had been incubated at 25 °C. The pathogen was inoculated 30 d after planting (DAP) using a spore suspension of *C. spinaciae* that contained approximately 1.0 × 10^6^ conidia per mL. The numbers of spores in the solution were determined using a direct microscopic counting procedure known as a Petroff-Hausser counting chamber (XB-K-25, Shiya Science and Technology Company, Guangzhou, China). There were 30 pots each of *C. spinaciae*-infected (C^+^) and *C. spinaciae*-uninfected (C^−^) plants. A suspension (20 mL) was sprayed onto the C^+^ plant (per pot), and the same volume of sterilized water was used for the C^−^ plants. All the plants were covered with a black plastic bag for 48 h, so they would be moist [37]. The incidence and DI of anthracnose were measured and counted three times at 35, 40, and 45 DAP to determine the success of inoculation.

The severity of anthracnose disease was classified into six different levels based on the percentage area of diseased lesions on the individual leaves: Level 0, no symptoms; Level 1, ≤10%; Level 2, 10.1–25.0%; Level 3, 25.1–50.0%; Level 4, 50.1–75.0%; and Level 5, 75.1–100%. The following equation was used to calculate the DI:$$\mathrm{DI}=\frac{\sum \left(n\times l\right)}{N\times L}\times 100$$
where *l*, *L*, *n*, and *N* represent the level of severity, the highest severity level, the number of leaves for each severity level, and the total number of leaves, respectively.

Stage 2. Powdery mildew naturally infected all the plants at 46 DAP. The incidence and DI of powdery mildew were then measured three times at 46, 51, and 56 DAP. The number of total plants per pot and those diseased with powdery mildew were counted to calculate the incidence by dividing the number of diseased plants by the total number of plants. Moreover, all the leaves were counted to assess the DI of powdery mildew. The classification and methods of calculating disease severity were the same as those described in stage 1 for anthracnose, we used the percentage area of powdery mildew instead of diseased lesions on the individual leaves. 

The photosynthetic indicators, including the net photosynthetic rate, transpiration rate, stomatal conductance, and concentration of intercellular carbon dioxide, were determined every 8 d from 40 DAP onward (40, 48, and 56 DAP). A GFS-3000 portable photosynthetic measuring system (Heinz Walz GmbH, Effeltrich, Germany) was used to measure the photosynthetic indicators.

The plants in each treatment were mixed and sampled randomly before harvest. There were four replicates, and each had more than 2 g of fresh leaves. These fresh leaf samples were used to measure the activities of superoxide dismutase (SOD), polyphenol oxidase (PPO), peroxidase (POD), and catalase (CAT). Approximately 0.1 g of fresh leaf sample was required for each enzyme assay as described by Li et al. [38]. Coomassie brilliant blue and uranine fluorescence were used to observe the powdery mildew mycelia as described by Wang et al. [39].

The aboveground parts of the plants in every pot were obtained at 60 DAP using scissors close to the soil and separated into different envelopes, dried, and weighed. The fresh weight of each pot was determined immediately, and the dry weight was determined after drying to a constant weight.

### 2.4. Statistical Analysis

The data are presented as the mean ± SE. SPSS 20.0 (IBM, Inc., Armonk, NY, USA) was used to analyze all the data. A two-way analysis of variance (ANOVA) was used to assess the treatment effects on the incidence, DI, photosynthetic indices, and biomass production with 10 replicates, except for the defense enzyme activities, which had four replicates. Independent-sample *t*-tests were used to determine significant differences between the C^+^ and C^−^ plants in all the parameters and defined at the 95% confidence level. A Spearman’s correlation heatmap was plotted using Origin 2022 (OriginLab, Northampton, MA, USA).

## 3. Results

### 3.1. Plant Fresh Weight and Dry Weight

The fresh and dry weights did not differ significantly between the C^−^ and C^+^ plants (*p* > 0.05) (Figure 1). The fresh weight was lower in the SFS than in the FS (relative decrease of 18.50%) (*p* < 0.05) (Figure 1a). In contrast, the dry weight was not significantly different across all the treatments (*p* > 0.05) (Figure 1b), which indicated that the soil microorganisms had a positive effect on the absorption of water by common vetch.

### 3.2. Disease Incidence and Disease Index

The *Colletotrichum spinaciae*-uninfected plants (C^−^) did not have anthracnose symptoms, while the C^+^ plants showed typical symptoms 5 d after inoculation with the pathogen. The disease incidence of common vetch anthracnose increased daily and reached 86.25~92.50% (Figure 2a). The DI showed a pattern similar to that of the disease incidence and reached 28.98~31.08 (Figure 2b). The DI and disease incidence did not differ significantly among the different soil treatments (*p* > 0.05).

Powdery mildew occurred at 46 DAP. The incidence of powdery mildew increased over time. Moreover, the incidence was significantly higher in the C^−^ plants (92.08~98.33%) than in the C^+^ plants (24.17%~55.83%) (*p* < 0.05) (Figure 3a). However, the incidence of powdery mildew did not differ significantly among the different soils under the same treatment (*p* > 0.05). The DI of powdery mildew was lower in the C^+^ plants (1.81~8.09) than in the C^−^ plants (12.85~31.84) (*p* < 0.05) (Figure 3b). The soil treatment had no impact on the DI of powdery mildew in both the C^+^ and C^−^ plants (*p* > 0.05).

### 3.3. Photosynthetic and Physiological Indices

The net photosynthetic rate was higher in the C^−^ plants than in the C^+^ plants by 156.06%~359.30% at 40 DAP (before powdery mildew occurrence) (*p* < 0.05). However, the net photosynthetic rate was lower in the C^−^ plants than in the C^+^ plants by 6.93~18.01% at 56 DAP (after powdery mildew infection for 10 d) (*p* > 0.05). The effect of powdery mildew alone was similar to the combined effect of anthracnose and powdery mildew infections (Figure 4a).

Anthracnose and powdery mildew significantly affected the plant transpiration rate. For example, the transpiration rate was significantly lower in the C^+^ plants than in the C^−^ plants (relative difference: 87.18~293.43%) at 40 DAP (*p* < 0.05) (Figure 4b).

Anthracnose and powdery mildew significantly affected the plant stomatal conductance. The stomatal conductance was significantly higher in the C^−^ plants than in the C^+^ plants by 90.63%~312.63% at 40 DAP (*p* < 0.05). The average stomatal conductance was slightly higher in the C^−^ plants than in the C^+^ plants by only 3.01% after powdery mildew infection for 10 d (*p* > 0.05) (Figure 5a).

Anthracnose and powdery mildew did not significantly affect the intercellular carbon dioxide (CO_2_) concentration of common vetch. The C^−^ and C^+^ plants had different intercellular CO_2_ concentrations on 40 DAP and 56 DAP (relative decrease; 12.49% and 12.94%, respectively). In contrast, the intercellular CO_2_ concentration did not differ significantly across the treatments (*p* > 0.05) (Figure 5b).

### 3.4. Enzyme Activities Related to Plant Defense

The enzymes related to defense in the C^+^ and C^−^ plants were determined after powdery mildew infection. The activities of SOD, PPO, and CAT were higher in the C^+^ than in the C^−^ plants (relative increase; 147.88–148.21%, 182.50–196.25%, and 98.81–264.91%, respectively) (*p* < 0.05) (Figure 6a,b,d). The activity of POD increased significantly in the C^+^ plants compared with the C^−^ plants in the FS (*p* < 0.05), while there was no significant difference in the SFS (*p* > 0.05) (Figure 6c). The values of the activities of SOD, PPO, and CAT were higher in the SFS than in the FS, which indicated that soil microorganisms had a negative impact on the activities of the plant defense enzymes (*p* < 0.05).

### 3.5. Correlation Analysis between the Level of Infection and the Photosynthetic Index, Biomass, and Defense Enzymes

There was a significant positive correlation between the disease incidence and DI for both anthracnose and powdery mildew (*p <* 0.05). There was no significant correlation between the disease incidence and DI of anthracnose and powdery mildew (*p >* 0.05). The disease incidence and DI of powdery mildew significantly positively correlated with the dry weight (*p <* 0.05) and significantly negatively correlated with the activities of POD, PPO, CAT, and SOD (*p <* 0.05). In addition, the disease incidence of powdery mildew significantly positively correlated with the intercellular carbon dioxide concentration (ICDC) (*p <* 0.05) (Figure 7).

## 4. Discussion

The interactions between synergistic diseases, such as anthracnose and powdery mildew, can lead to unexpected dynamics in plant microbial multi-pathogen systems. In this study, the response of common vetch infected with *C. spinacia* to the natural development of powdery mildew was analyzed using a greenhouse experiment. The plants infected with *C. spinacia* were less likely to be infected with powdery mildew. Moreover, the plants that were first infected with *C. spinacia* had a lower severity of powdery mildew symptoms, with a higher activity of defense-related enzymes (SOD, PPO, POD, and CAT) involved in induced systemic resistance by anthracnose. Our research confirmed the pre-infection interactions of the anthracnose pathogen to be “ecological interference,” which reduced the contact rate and development to secondary infection by powdery mildew. In addition, the damage caused by powdery mildew infection alone and that caused by the combined infection with anthracnose and powdery mildew were not significantly different.

The photosynthetic indices were lower in the C^+^ plants than in the C^−^ plants before powdery mildew infection, which indicated that anthracnose decreases the photosynthesis of common vetch. Anthracnose may affect photosynthesis by reducing the contents of chlorophyll a and b and inhibiting the PSII photochemical activity [18]. Castro et al. found that the *Colletotrichum* sp. decreases light capture and the assimilation of CO_2_ in the mesophyll, thus decreasing the photosynthetic performance [40]. Additionally, lower gaseous exchanges in anthracnose-infected plants could be owing to the reduction in leaf pigments, thus decreasing photosynthesis [41].

The photosynthetic indices of the C^+^ and C^−^ plants were approached gradually after powdery mildew infection, indicating that powdery mildew suppresses the photosynthesis of common vetch. Watanabe et al. found that severe infection with powdery mildew significantly reduced the photosynthetic activity of the sprouts [42]. Powdery mildew reduces photosynthesis by reducing the green leaf area through its visual lesions and influencing the gas exchange of the remaining green leaf tissues [43]. Herein, there were interactions between powdery mildew and anthracnose in common vetch, which could also be affected by multiple fungal pathogens. However, Lopes and Berger (2001) showed no interaction between rust and anthracnose [44], which indicates the diversity of plant responses to infection with multiple pathogens.

The disease incidence and DI of powdery mildew differed significantly between the C^+^ and C^−^ plants, which indicated a significant interaction between the two diseases mediated by common vetch under greenhouse conditions. The C^+^ plants infected with pathogens were less susceptible to powdery mildew than the C^−^ plants, which indicates that common vetch is more tolerant to powdery mildew after *C. spinaciae* infection. Similarly, Harth et al. found that the infection of powdery mildew decreased in *Cucurbita pepo* after inoculation with Zucchini yellow mosaic virus [30]. However, the soil did not induce significant variation in the development of powdery mildew. Moreover, the leaves of common vetch that had been pre-infected with anthracnose had substantially fewer powdery mildew mycelia, possibly owing to the necrotic reactions of the plant hypersensitive response. Pre-infection with *C. spinaciae* causes the death of plant tissue, which reduces the sites of pathogen infection and the absorption of carbon from plants [45], which was similar to the microscopic observations (Appendix A).

Plant diseases can cause serious losses in yield. For example, anthracnose causes serious losses in the yield of velvetleaf (*Abutilon theophrasti*) [46], Chinese goldthread (*Coptis chinensis*) [37], alfalfa (*Medicago sativa*) [47], plum (*Pyrus* spp.) [48], qing qian liu (*Cyclocarya paliurus*) [49], and other crops in China. Similarly, powdery mildew reduces the production of *Euonymus japonicas* [50] and wheat [51] spiny bitter gourd (*Momordica cochinchinensis*) [52], and Jerusalem artichoke (*Helianthus tuberosus*) [53]. Notably, powdery mildew caused by *Erysiphe pisi*, *Erysiphe trifolii*, and *Erysiphe baeumleri* can cause 25–50% yield losses [54]. A previous study found that infection with anthracnose can significantly decrease the growth and biomass of common vetch [37]. Herein, the biomass of the aboveground fresh and dry weights was not significantly different between the C^+^ and C^−^ plants after infection with powdery mildew, which confirmed the strong interaction between anthracnose and powdery mildew. The infection of powdery mildew on common vetch is reduced by anthracnose infection. Generally, the leaf abrasions caused by inoculation can provide an easier avenue for the fungus to invade the plants [30]. However, powdery mildew ingress seemed to be greatly reduced after anthracnose had already caused damage to common vetch in our study. The two diseases caused the same damage as anthracnose alone, which suggested that anthracnose could use some mechanism to prevent powdery mildew invasion. Our hypothesis was partially upheld.

Plant defense enzymes strongly respond to pathogens [55]. The damage to the plant leaves caused by pathogens leads to the formation of reactive oxygen species (ROS) [56], which play a key role in plant defense. The accumulation of ROS can damage the plant tissue [57]. Plants produce antioxidative enzymes, such as SOD, CAT, POD, and PPO, to maintain the homeostasis of ROS by scavenging redundant ROS [58]. In this study, the activities of enzymes related to defense were significantly higher in the C^+^ plants than in the C^−^ plants. Moreover, infection with anthracnose upregulated the defense genes and produced several enzymes related to defense [37,59,60,61,62].

Anthracnose induced a strong defense response and increased the activities of enzymes related to defense, thus inhibiting the growth of powdery mildew. This explains why powdery mildew was more severe in the C^−^ plants than in the C^+^ plants. Agents derived from plants induce resistance to powdery mildew by activating the biochemical defense of the treated leaves, such as by increasing the activities of PPO and POD [63]. This was further supported by an analysis based on the Spearman’s correlation matrix of the activities of the plant defense enzymes and the severity of powdery mildew. Chitosan and potassium reduced the severity of powdery mildew and increased the activities of PPO, POD, chitinase, and β-1,3-glucanase in okra (*Abelmoschus esculentus*) plants [64]. This mechanism related to plant defense enzymes has been used to develop resistance to plant powdery mildew. For example, an extract of the brown alga *Ascophyllum nodosum* significantly regulated the activity of defense-related enzymes, such as CAT, POD, and PPO, to induce the active defense of squash (*Cucurbita moschata*) [65], strawberry (*Fragaria × ananassa*) [23], and cucumber (*Cucumis sativus*) against powdery mildew [66].

The defense response in the C^+^ plants induced by anthracnose may also lead to the production of some secondary metabolites related to defense, such as phenolics and flavonoids [26]. These metabolites and enzymes related to defense can improve the resistance to powdery mildew. Moreover, this has been confirmed at the physiological, biochemical, and genetic levels. Moreover, the levels of expression of the genes related to hormone signaling, POD, and cell wall degradation under powdery mildew infection are higher in the resistant varieties than in the susceptible varieties [67].

The resident soil microbiota did not affect the occurrence of anthracnose and the subsequent natural development of powdery mildew. Equally, our hypothesis that common vetch grown with resident soil microorganisms would have less severe disease was not supported. The plants grown in unsterilized soil had higher fresh weights and lower activities of plant defense enzymes, which suggested that the soil microorganisms had a positive effect on the absorption of water by plants and plant defense to diseases. Recent research has pinpointed the concept that enhanced nutrient uptake, improved root architecture, and protection of the host against biotic and abiotic stress are key functions of the soil microbiome [68]. Further research is required to clarify the microbial mechanism in complex plant–fungal pathogen interactions.

In conclusion, this study showed that anthracnose infection can establish “ecological interference”, thus reducing the contact rate and development of powdery mildew infection. The defense response and increase in the activities of defense-related enzymes, including SOD, PPO, POD, and CAT, induced by anthracnose infection can thus improve the plant immune response to powdery mildew. The physiological and biochemical changes confirmed the interactive effect between anthracnose and powdery mildew in common vetch. The resident soil microbiota from the growth of common vetch in the field had no significant impact on the severity of plant disease, but they increased the activities of the enzymes related to plant defense and showed a potential function to improve plant disease resistance.

## Figures and Tables

**Figure 1 plants-13-00052-f001:**
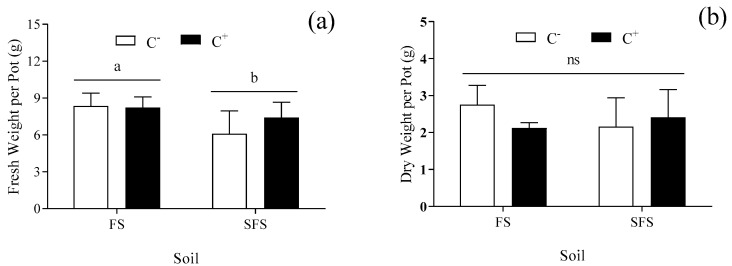
Fresh (**a**) and dry (**b**) weight of common vetch—uninfected (C^−^) vs. infected (C^+^) with *Colletotrichum spinaciae* grown in field soil (FS) and sterilized field soil (SFS). Different lowercase letters above the bars indicate a significant difference between the FS and SFS at the *p* < 0.05 level. ns means there is no significant difference between FS and SFS.

**Figure 2 plants-13-00052-f002:**
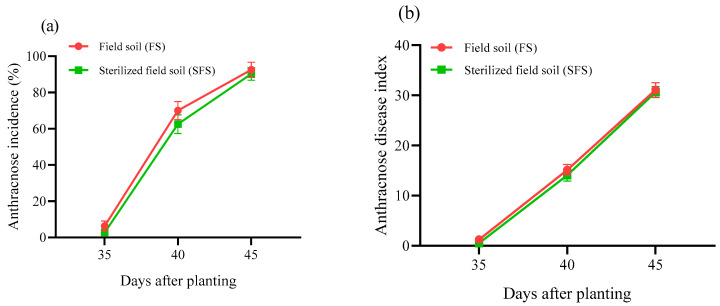
Anthracnose incidence (**a**) and disease index (**b**) of common vetch grown in field soil (FS) and sterilized field soil (SFS).

**Figure 3 plants-13-00052-f003:**
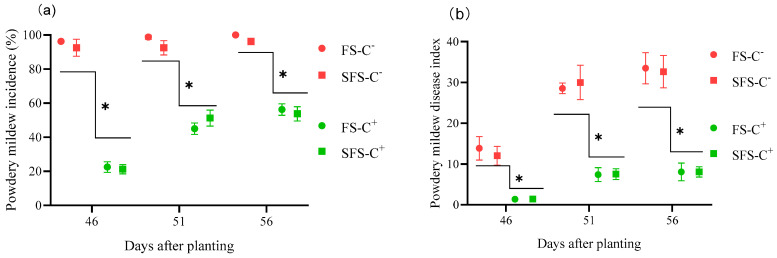
Powdery mildew incidence (**a**) and disease index (**b**) in common vetch uninfected (C^−^) and infected (C^+^) with *Colletotrichum spinaciae* grown in field soil (FS) and sterilized field soil (SFS). ***** significant difference between common vetch infected and uninfected with *C. spinaciae* at the *p* < 0.05 level on the same date.

**Figure 4 plants-13-00052-f004:**
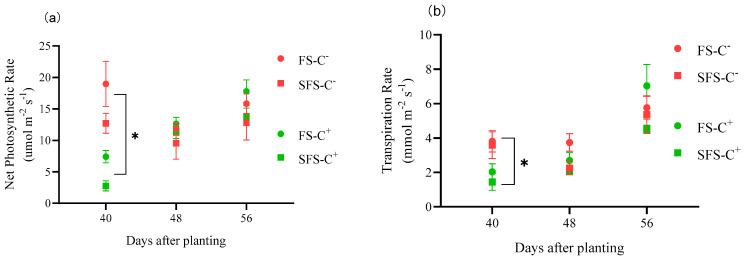
Net photosynthetic rate (**a**) and transpiration rate (**b**) in common vetch uninfected (C^−^) and infected (C^+^) with *Colletotrichum spinaciae* grown in field soil (FS) and sterilized field soil (SFS). ***** significant difference between common vetch infected and uninfected with *C. spinaciae* at the *p* < 0.05 level on the same date.

**Figure 5 plants-13-00052-f005:**
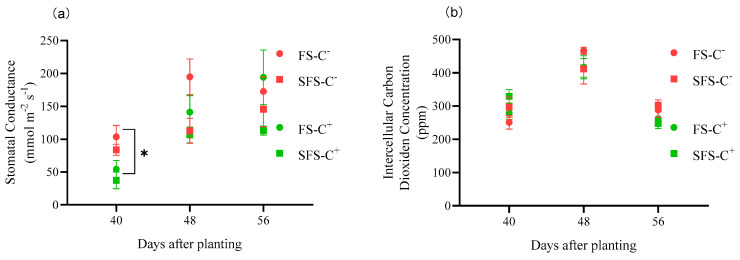
Stomatal conductance (**a**) and intercellular carbon dioxide concentration (**b**) in common vetch infected and uninfected with *Colletotrichum spinaciae* grown in field soil (FS) and sterilized field soil (SFS). ***** significant difference between common vetch infected and uninfected with *C. spinaciae* at the *p* < 0.05 level on the same date. The soil treatment had no impact on the photosynthetic and physiological indices of common vetch when exposed to anthracnose and powdery mildew (*p* > 0.05).

**Figure 6 plants-13-00052-f006:**
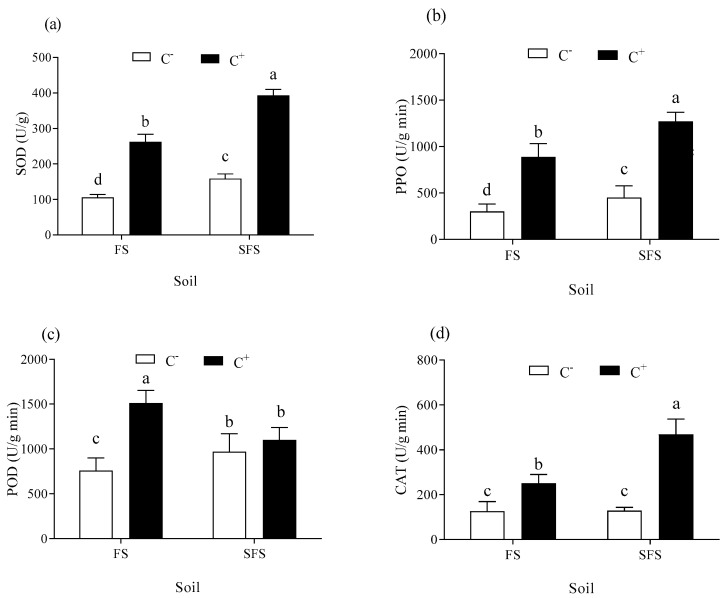
Defense-related enzyme activities of SOD (**a**), PPO (**b**), POD (**c**), and CAT (**d**) in common vetch plants uninfected (C^−^) vs infected (C^+^) with *Colletotrichum spinaciae* grown in field soil (FS) and sterilized field soil (SFS). Different lowercase letter above the bars indicates a significant difference between common vetch infected and uninfected with *C. spinaciae* grown in field soil (FS) and sterilized field soil (SFS) at the *p* < 0.05 level. CAT, catalase; POD, peroxidase; PPO, polyphenol oxidase; SOD, superoxide dismutase.

**Figure 7 plants-13-00052-f007:**
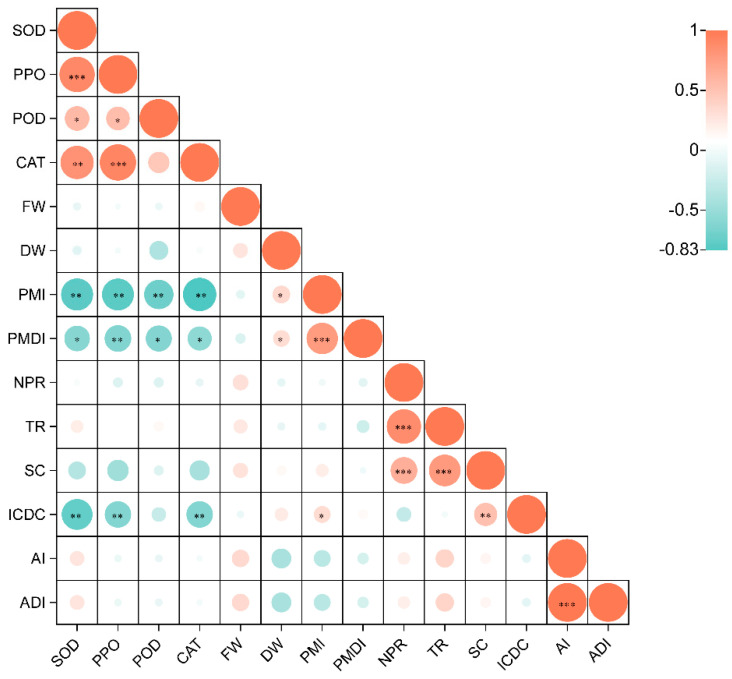
Heatmap based on the Spearman’s correlation matrix of the biomass of fresh weight (FW) and dry weight (DW), and the activities of POD, PPO, CAT, and SOD at harvest. The profiles before harvest include the photosynthetic index of net photosynthetic rate (NPR) and stomatal conductance (SC), transpiration rate (TR) and intercellular carbon dioxide concentration (ICDC), the anthracnose incidence (AI) and anthracnose disease index (ADI), and the powdery mildew incidence (PMI) and powdery mildew disease index (PMDI). CAT, catalase; POD, peroxidase; PPO, polyphenol oxidase; SOD, superoxide dismutase. * statistical difference the two variables that correspond to the small grid at the level of 0.05 (0.01 < *p* ≤ 0.05). ** statistically significant difference at the level of 0.01 (0.001 < *p* ≤ 0.01). *** extremely significant statistical difference at the level of 0.001 (0.0001 < *p* ≤ 0.001).

## Data Availability

Data are contained within the article and Appendix A.

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
