# Peer review of "Prior Infection by Colletotrichum spinaciae Lowers the Susceptibility to Infection by Powdery Mildew in Common Vetch"

_plants, 2023, doi:10.3390/plants13010052_

Round 1

Reviewer 1 Report

Comments and Suggestions for Authors

Methodology

The “naturally occurrence” of powdery mildew in the experiment makes difficult to quantify its homogeneity in plants, besides, to replicate the experiment the part of naturally occurrence is difficult to replicate as no control of the infection was carried out in time and quantity.

Conclusion

 The conclusion is not in consonance with data “In conclusion, powdery mildew causes a lower disease incidence and severity in common vetch than C. spinaciae infection”

Lines 204 and 205 “Moreover, the incidence of powdery mildew was significantly higher in the C- plants than in C+ plants (92.08%~98.33% and 24.17%~55.83%, respectively) (P<0.05)”

What is written in the conclusion does not match the results. What is clearly is that anthracnose infection decreased the incidence of powdery mildew.

Conclusion needs revision.

Days after planted or dap. Use in text an figures “dap”.

See comments in MS.

Comments on the Quality of English Language

Minor editing of English language required

Author Response

We acknowledge the reviewer’s many supportive comments and recommendations, which we found very helpful for revising the manuscript. We have revised the Title, the Introduction, and Materials and Methods, particularly completed the analytical section by performing a Spearman's rank correlation analysis between the level of infection and photosynthetic indexes, biomass production and defence enzymes. We also revised Discussion and rewritten the conclusion, added new references. We have marked the major changes with yellow highlight. The detailed response to the reviewer is as follows.

Reviewer 2 Report

Comments and Suggestions for Authors

A review of the manuscript titled "Colletotrichum spinaciae Infection Reduces the Subsequent Powdery Mildew to Common Vetch (Vicia sativa)" by Li et al. I found the manuscript interesting and valuable. However, I have some important comments. I therefore recommend a comprehensive revision. See below for details.

Main comments:

The two pathogens have different effects on pea plants, which have different lifestyles. Colletotrichum fungi are classified as hemibiotrophic pathogens, while Erysiphe pisi is biotrophic. This aspect is not addressed in the paper. This requires a fundamental change in the approach to the research topic and a change in the literature review. I have doubts about the correctness of the authors' approach to the results and conclusions presented in the paper. Powdery mildew occurred naturally on the plants and it is not certain that the plants of different experimental variants were infected in the same way. It is necessary to change the title of the paper and point out that powdery mildew has developed naturally. This information should be included in the summary of the article.

Please complete the analytical section by performing a Spearman's rank correlation analysis between the level of infection and photosynthetic indexes, biomass production and defence enzymes.

Specific comments:

L15 Which soil microorganisms? Please explain this aspect in your working method and results.

L106. What is "Lanjian No. 2"?

L108-111. State the name of the strain and the place where the isolate of C. spinaciae is kept (collection).

L143. How was the spore dilution calculated?

Author Response

(The authors gave the same response as above.)

Round 2

Reviewer 2 Report

Comments and Suggestions for Authors

The work has been revised. All commentaries have been taken into account. I recommend that the work be printed in its present form. I congratulate the authors on their well-conducted and interesting research.

Author Response

Many thanks for your  constructive comments, we deeply appreciate your help.

Best wishes!

Tingyu